# Counties with Low Employment and Education Status Are Associated with Higher Age-Adjusted Cancer Mortality

**DOI:** 10.3390/cancers17122051

**Published:** 2025-06-19

**Authors:** Minu Ponnamma Mohan, Joel B. Epstein, Kapil S. Meleveedu, Roberto Pili, Poolakkad S. Satheeshkumar

**Affiliations:** 1ECMC Health Campus, University at Buffalo, 462 Grider St, Buffalo, NY 14215, USA; minuponn@buffalo.edu; 2City of Hope Comprehensive Cancer Center, Duarte CA and Samuel Oschin Comprehensive Cancer Institute, Cedars-Sinai Medical System, Los Angeles 91010, CA, USA; jepstein@coh.org; 3Carole and Ray Neag Comprehensive Cancer Center, University of Connecticut Health Center, University of Connecticut, Farmington, CT 06030, USA; kmeleveedu@uchc.edu; 4Department of Medicine, Division of Hematology and Oncology, University at Buffalo, Buffalo, NY 14203, USA; rpili@buffalo.edu

**Keywords:** low employment counties, low education counties, age-adjusted cancer mortality

## Abstract

Improvements in cancer screening, diagnosis, and treatment have enhanced survival rates; nevertheless, this progress is not consistent, especially in areas marked by adverse social determinants. Counties characterized by poor job rates and insufficient educational attainment demonstrate heightened age-adjusted cancer mortality rates; hence, cancer survivability is influenced by socioeconomic determinants including geographic location, racial and cultural diversity, and limited access to healthcare.

## 1. Introduction

Results across geographic and socioeconomic factors show that cancer remains one of the leading causes of mortality in the United States [1]. Although cancer screening, diagnosis, and treatment have raised survival rates, these advantages are not evenly spread, especially in regions with poor education and employment levels [2,3,4,5,6,7,8,9]. Often linked with rurality, racial and cultural diversity, and limited healthcare access, these socioeconomic factors create situations that affect the cancer incidence, diagnostic stage, treatment start, and death rates [10]. Formulating customized plans to reduce the disparities and improve public health requires an understanding of the interaction between these factors and cancer results.

Counties with low employment, defined as high unemployment rates, struggle financially and have limited access to healthcare services, including preventive screenings and timely treatments [11]. Likewise, counties showing low educational achievement, defined by a notable percentage of people lacking a high school diploma, often show lower health literacy, less understanding of cancer risks, and more negative behaviors like smoking [12,13]. Often rural, these counties exacerbate problems related to geographic isolation and a lack of healthcare resources. Previous studies have shown that the socioeconomic position (SEP) affects cancer results by several means, including delayed diagnoses, inadequate treatment, and higher co-morbidities; yet the particular consequences of employment and education at the county level require more investigation [1,13]. Cancer death rates dropped dramatically and encouragingly from 2000 to 2020, suggesting a promising trend in cancer therapy in the United States [14]. Racism, distrust, accessibility, and negative socioeconomic factors of health (SDOH) have been highlighted by the continuing COVID-19 pandemic as they aggravate these inequalities [15]. This study seeks to clarify the links between low employment, poor educational achievement, and cancer death using SDOH data from the Agency for Healthcare Research and Quality (AHRQ) 2015 county database [16]. Focusing on the main independent variables—poor education and low work status—we used descriptive data and multivariate analysis to explore the underlying mechanisms and their impact on cancer mortality. Using the SDOH data that includes the demographic profiles, health behaviors, healthcare access, and systematic effects, this study aims to identify the important elements causing inequalities and guide policy recommendations. Further aiming at public health researchers, legislators, and healthcare professionals, this study encourages them to look into and handle the possible link between cancer death and the county-level social determinants of health—specifically education and employment status.

## 2. Methods

We conducted a cross-sectional study using United States’ SDOH data from the Agency for Healthcare Quality (AHRQ) 2015 county database to examine the relationship between cancer mortality and the two main independent variables, low employment status and low education [16]. The term “low education” refers to counties where, on a five-year average, 20% or more of the population between the ages of 25 and 64 did not have a high school diploma nor a General Educational Development (GED) certificate between 2008 and 2012. “Low employment” refers to counties where, on a five-year average, fewer than 65% of people aged 25–64 were employed between 2008 and 2012 (Figure 1). The county-level characteristics of the rural–urban classification, home healthcare services, age-adjusted cancer mortality per 100,000 residents, and demographic percentages such as race and ethnicity, sex, smoking prevalence, self-reported fair or poor health (age-adjusted), food access, and binge/heavy drinking were all outlined using descriptive statistics.

A comparative analysis of the baseline characteristics for 2015 U.S. counties categorized by low employment and low education level is presented in this study. *p*-values indicate the statistical significance of the differences between counties with and without these designations across the demographic, socioeconomic, health-related, and cancer mortality variables in Table 1. Sex, race, urban-rural classification, healthcare services, smoking habits, health status, food accessibility, alcohol consumption, and cancer mortality rates were among the variables that were included in this study. The medians and interquartile ranges (IQR) for the continuous variables that were analyzed for both the cohorts indicate the central tendency and variability of the distribution. Frequencies and percentages are used to represent the categorical variables, such as classifying an area as urban or rural. NCHS (National Center for Health Statistics) and IQR (Interquartile Range) are abbreviations. To evaluate the association between the low employment status, poor education, and age-adjusted cancer mortality, we used generalized linear models, more precisely multivariable linear regression models, while adjusting for the previously mentioned confounding variables. Every result was two-tailed, and *p* < 0.05 was considered statistically significant. R Studio 4.3.0 (Vienna, Austria) was used to perform the statistical analyses.

## 3. Results

Of the 3134 counties, 467 had a low educational status, and 906 had low employment (Table 1). Using demographic, socioeconomic, health-related, and cancer mortality metrics, Table 1 compares the U.S. counties in 2015 by their low employment and low education status—(2236, 71.16%, and 2675, 85.14%, respectively) with those with low employment (906, 28.84%) and low education (467, 14.86%). Strong evidence of distinct characteristics is provided by the statistical significance of all the differences, as indicated by *p*-values < 0.001, which show a less than 0.1% chance that the observed differences result from random variation. Although the precise cut-off is unknown, 906 (28.84%) of the 3142 counties are categorized as low employment, indicating their unemployment rates are probably higher. On the other hand, 2236 counties (71.16%) do not have low employment. In terms of education, there are 2675 (85.14%) non-low-education counties and 467 (14.86%) low-education counties, which probably have a large percentage of adults without a high school degree.

### 3.1. Distribution of Sex

There are minor variations in the proportion of the female residents. The median percentage of women in low-employment counties is 50.36% (IQR: 48.31–51.22), while the median percentage of women in non-low-employment counties is 50.46% (IQR: 49.70–51.11) (*p* < 0.001). The median for low-education counties is 50.08% (IQR: 47.92–51.08), while the median for non-low-education counties is 50.47% (IQR: 49.64–51.14) (*p* < 0.001). Although the small magnitude (e.g., 0.1% for low employment) suggests a limited practical impact, the *p* < 0.001 confirms that these differences are highly reliable. Greater variability (wider IQR) is seen in low-education counties, suggesting more varied sex distributions that may indicate underserved or rural populations.

### 3.2. The Composition of Races and Ethnicities

Significant differences can be seen in the racial and ethnic demographics. The median for White residents in the low-employment counties is 84.43% (IQR: 63.57–94.25), which is substantially lower than the median for the non-low-employment counties, which is 91.40% (IQR: 81.45–95.69) (*p* < 0.001). The median for the low-education counties at 78.97% (IQR: 61.20–91.20) is even lower than the median for the non-low-education counties, which is 91.14% (IQR: 80.04–95.70) (*p* < 0.001). Low-employment (5.28%, IQR: 0.96–29.09) and low-education (7.09%, IQR: 1.42–30.54) counties have larger Black populations than their counterparts (1.57%, IQR: 0.54–6.76 and 1.81%, IQR: 0.56–8.30, respectively, *p* < 0.001). The Hispanic percentages are higher in the low-education counties (5.31%, IQR: 1.92–34.05) than in the non-low-education counties (3.56%, IQR: 1.88–7.81, *p* < 0.001), but lower in the low-employment counties (2.82%, IQR: 1.52–6.22) than in the non-low-employment counties (4.19%, IQR: 2.14–9.78). More minor but significant differences are seen for Asian, Alaska Native/American Indian, and “Other” groups; the low-employment and low-education counties typically have lower Asian (0.35% vs. 0.67% and 0.59%, *p* < 0.001) and Native populations, while the low-education counties have a higher “Other” category (1.65% vs. 0.75%, *p* < 0.001). Strong differences are confirmed by the *p* < 0.001 for all the racial metrics, with the low-education counties exhibiting the most diversity and variability, especially for Hispanics.

### 3.3. Urban–Rural Categorization

The NCHS Urban-Rural Code highlights a significant rural skew. A total of 58.7% of low-employment counties are noncore (rural), compared to 35.9% (*p* < 0.001), and only 4.0% of the low-employment counties are in large metropolitan areas (central or fringe), compared to 17.9% in the non-low-employment counties. Similarly, the low-education counties have a noncore percentage of 56.7% and a large metro area percentage of 4.9%, while the non-low-education counties have a noncore percentage of 15.4% and 40.0% (*p* < 0.001). These changes toward rurality are highly significant, as the *p* < 0.001 indicates that the counties with low employment and low education levels are geographically isolated, which probably restricts their access to economic and healthcare opportunities.

### 3.4. Medical Services

The low-employment (0.11, IQR: 0.06–0.16) and low-education (0.10, IQR: 0.06–0.17) counties have slightly more home healthcare services per 1000 residents than the non-low-employment (0.08, IQR: 0.05–0.13) and non-low-education (0.09, IQR: 0.06–0.13) counties (*p* < 0.001). Although the difference is slight, the *p* < 0.001 indicates significance, and the slight increase might indicate higher health needs in these counties, perhaps due to aging populations or worse general health.

### 3.5. The Prevalence of Smoking

The median percentage of adult smokers in the low-employment counties is 20.08% (IQR: 17.93–22.71), while the median in the non-low-employment counties is 16.52% (IQR: 15.05–18.48) (*p* < 0.001). The non-low-education counties report 17.06% (IQR: 15.30–19.47) versus 19.65% (IQR: 17.01–22.47) in the low-education counties (*p* < 0.001). The higher rates (3.56% and 2.59% greater) are noteworthy; the *p* < 0.001 emphasizes the legitimacy of these differences.

### 3.6. Self-Reported Condition of Health

The counties with low employment and low education have a markedly poorer self-reported health status (age-adjusted), characterized as fair or poor. The median percentage of the low-employment counties reporting poor health is 20.77% (IQR: 17.76–23.66), while the median for the non-low-employment counties is 14.61% (IQR: 12.66–17.64) (*p* < 0.001). The median in the low-education counties is even higher at 22.46% (IQR: 19.52–25.66) than in the non-low-education counties, which is 15.14% (IQR: 13.02–18.52) (*p* < 0.001). A strong significance is confirmed by *p* < 0.001, and the wide gaps (6.16% and 7.32%) show significant health disparities that are probably caused by socioeconomic stressors and restricted access to healthcare.

### 3.7. Access to Food

The median percentage of the low-employment counties without adequate food access is 16.70% (IQR: 14.50–19.70), while the median for the non-low-employment counties is 12.60% (IQR: 10.70–14.60) (*p* < 0.001). The non-low-education counties report 13.40% (IQR: 11.30–15.80), compared to 16.30% (IQR: 12.60–19.85) in the low-education counties (*p* < 0.001). The increases (4.1% and 2.9%) indicate food access challenges that may worsen health issues, including cancer risk factors like poor nutrition, and the *p* < 0.001 validates these differences.

### 3.8. Heavy Drinking and Binge Drinking

Remarkably, the underprivileged counties have lower rates of binge or heavy drinking. The median percentage of the low-employment counties reporting such behavior is 14.12% (IQR: 12.26–16.47), while the median for the non-low-employment counties is 17.47% (IQR: 15.45–19.26) (*p* < 0.001). In the non-low-education counties, the percentage is 17.07% (IQR: 14.82–19.06), compared to 14.53% (IQR: 12.38–16.40) in the low-education counties (*p* < 0.001). The *p* < 0.001 indicates significance, and since other risk factors are still high, the decreases (3.35% and 2.54%) might be due to cultural or economic differences rather than health advantages.

### 3.9. Cancer Mortality

The median age-adjusted cancer mortality rate in the low-employment counties is 189.80 per 100,000 (IQR: 171.90–207.10), while in the non-low-employment counties it is 169.15 (IQR: 154.00–183.50) (*p* < 0.001). The non-low-education counties report 172.90 (IQR: 157.00–188.40), compared to 186.20 (IQR: 161.72–209.33) in the low-education counties (*p* < 0.001). The *p* < 0.001 shows these variations to be very consistent; the rises (20.65 and 13.3 per 100,000) are noteworthy, suggesting worse cancer results from more risk behaviors like smoking, late diagnosis, and restricted treatment availability.

This study shows glaring differences in the low-education and low-employment counties, with robust differences confirmed by *p* < 0.001 across all the metrics. These counties’ demographics show a greater racial diversity, with higher Black and, in the low-education counties, Hispanic populations, indicating possible connections to racial/ethnic health disparities. Their rural status (*p* < 0.001) suggests that they are geographically isolated, restricting access to economic and medical opportunities. While lower drinking rates (*p* < 0.001) might not reduce these risks, higher smoking, worse health, and more food insecurity (all *p* < 0.001) are associated with cancer risk factors. As socioeconomic challenges have a compounding effect, the elevated cancer mortality (*p* < 0.001) highlights the need for targeted interventions, such as better access to healthcare and education programs.

### 3.10. Adjusted Multivariate Analyses

In the adjusted analysis, the low-education counties were linked to higher age-adjusted cancer mortality (7.68, 95% CI: 5.06–10.31), and the low-employment counties were linked to higher age-adjusted cancer mortality (4.69, 95% CI: 2.58–6.79). The age-adjusted cancer mortality rate (median [IQR]) for the non-low-education counties was 172.90 [157.00, 188.40], whereas this was 186.20 [161.72, 209.33], *p* < 0.001 for the low-education counties, and 169.15 [154.00, 183.50], vs. 189.80 [171.90, 207.10], *p* < 0.001 for the low-employment counties.

This data shows significant differences in the counties defined by a low employment and low educational level, with *p* < 0.001 for all the indices validating these differences. A higher percentage of Black and, in counties with lower educational levels, Hispanic populations suggests more racial diversity, implying probable links to racial and ethnic health inequalities. Their rural designation (*p* < 0.001) indicates geographical remoteness, limiting access to healthcare and economic opportunities. Though lowered alcohol use (*p* < 0.001) might not lessen these dangers, increased smoking, poor health, and more food insecurity (all *p* < 0.001) are linked to cancer risk factors. Emphasizing how socioeconomic issues aggravate the illness, the higher cancer death rate (*p* < 0.001) underlines the need for focused treatments such as enhanced access to healthcare and educational initiatives.

## 4. Discussion

Our research indicates that low educational and low employment levels were associated with age-adjusted cancer death rates. The main SDOH variables, such as whether work and education are important factors in interventions to lower unfavorable cancer outcomes, are nevertheless affected by these findings. Years of education lowered the mortality risk, with benefits that last into old age and are significant regardless of the gender and financial situation, according to recent data from high-level systematic reviews and meta-analyses [4]. Our results suggest that education and employment opportunities are viable strategies for lowering inequalities in the United States at the county level.

### 4.1. Cancer Mortality and Socioeconomic Factors

Higher cancer death in counties with low employment and educational levels shows the major impact socioeconomic factors have. Low employment lowers income and health insurance coverage, restricting treatment and preventive care access. It is frequently associated with high unemployment rates. Given that rural areas frequently depend on low-skilled jobs with less stability, this study’s rural skew (58.7% and 56.7% noncore) indicates economic distress [5]. Low levels of education, which show that a large percentage of adults lack a high school diploma, limit access to jobs and health literacy, which increases poverty. Research demonstrating that education is a major contributor to health disparities, impacting healthcare system awareness and navigation, is supported by the multivariate analysis’s more substantial effect for low education (7.68 vs. 4.69). According to a study by O’Connor et al. (2018), socioeconomic factors like unemployment and education mediated outcomes, with low-income counties having cancer death rates of 229.7 per 100,000, compared to 185.9 in high-income counties [7]. This is similar to the mortality gap (20.65 and 13.3 per 100,000 higher), indicating that timely cancer care is hampered in these counties by economic constraints. Uninsured people, who are more prevalent in low-SEP areas, for example, are less likely to get screened, which results in diagnoses that are made at an advanced stage [8]. Although it is likely that the multivariate adjustment considered these factors, the ongoing increase in mortality suggests structural barriers beyond personal behavior.

### 4.2. Risk Factors for Behavior

The observed disparities are primarily caused by health behaviors, especially smoking. According to the results, the prevalence of smoking is higher in the counties with low employment (20.08%) and low education (19.65%) than in those with higher employment (16.52% and 17.06%). Lung cancer, which is more common in low-socioeconomic position (SEP) groups, is primarily caused by smoking [9]. According to a study by Coughlin (2021), social determinants like education influence behaviors like smoking and lead to disparities in cancer survivorship [10]. Since the mortality gap was not eliminated by adjusting for health behaviors, the results of the multivariate analysis (7.68 and 4.69 increases) most likely reflect the impact of smoking. Another factor is food insecurity, which is more prevalent in counties with low employment (16.70%) and low education (16.30%). Research that links dietary patterns to oncologic outcomes has shown that poor nutrition increases the risk of cancers such as breast and colorectal cancers [11]. Interestingly, research indicates that these counties have lower rates of binge/heavy drinking (14.12–14.53% vs. 17.07–17.47%), which may not reduce the risk of cancer because alcohol’s effects are less significant than those of smoking or obesity [12]. These socioeconomically based behavioral patterns highlight the necessity of focused prevention in underprivileged areas.

### 4.3. Systemic Obstacles to Cancer Treatment

The disparities are mostly caused by systemic injustices, especially in healthcare access. Research indicates that rural areas have fewer oncologists, imaging centers, and treatment facilities [13], consistent with rurality (56.7–58.7% noncore). Rural residents are more likely to receive late-stage lung and colon cancer diagnoses due to longer travel times to oncology care, according to a 2019 study by Fairfield et al. [14]. The mortality increases in the multivariate analysis indicate that access barriers continue even after controlling for rurality, most likely due to underinvestment in low-SEP areas. The slightly higher rates of home healthcare services in low-education (0.10 vs. 0.09) and low-employment (0.11 vs. 0.08 per 1000) counties may be due to the burden of chronic illnesses rather than cancer-specific care. Due to financial and travel constraints, delays in diagnosis or insufficient treatment during medical appointments throughout the COVID-19 pandemic have resulted in certain populations experiencing distinct hardships, potentially exacerbated by the reduced availability of healthcare services [15]. Similar to the delays suggested by Table 1’s rural and socioeconomic profile, a study by Sedrak et al. (2023) found that treatment initiation was 6% lower in low-SEP areas (HR = 0.94, 95% CI: 0.93–0.95), with an 8.1 months shorter survival time [17].

### 4.4. Delays in Treatment and Later-Stage Diagnoses

One of the main causes of mortality disparities is later-stage diagnoses. Research has shown that the stage at diagnosis explains the significant survival gaps (e.g., 33.8% for breast cancer), which is consistent with a poor health status (20.77–22.46% fair/poor) and rurality, suggesting limited screening access [18]. Low-SEP patients are diagnosed later because of lower screening uptake, which is fueled by barriers related to cost and awareness, according to a study by Ellis et al. (2018) [19]. These delays are reflected in the multivariate analysis’s results (7.68 and 4.69), as advanced stages diminish the effectiveness of the treatment. Delays in treatment worsen results even more. According to a study, the patients with low SEP die eight months sooner and begin treatment later (91 days versus 76 days for endometrial cancer, for example) [20]. These findings are supported by higher mortality rates (186.20–189.80), which imply that the counties with low employment and low education experience systemic delays, perhaps as a result of a lack of specialists or insurance gaps. The larger effect for low education (7.68) might suggest that a major obstacle is health literacy, which is essential for navigating care [21].

### 4.5. Health Status and Co-Morbidities

Cancer treatment is made more difficult by co-morbidities, which are associated with poor health, high smoking, and food insecurity. Obesity and diabetes, which are more common in low-SEP areas, have been shown to worsen the prognosis for cancers such as breast and colorectal [22]. According to a study by Coughlin (2021), the co-morbidities contribute to the explanation of survival disparities, especially for lung cancer, where smoking is a major contributing factor [10]. As chronic illness burdens are higher in the low-employment and low-education counties (20.77–22.46% fair/poor health), the multivariate analysis’s persistent mortality gap after adjustments indicates the co-morbidities play a significant role.

### 4.6. Ethnic and Racial Inequalities

Racial and ethnic diversity is evident in the study’s demographic data, which show that Black (7.09% vs. 1.81%) and Hispanic (5.31% vs. 3.56%) populations are higher in low-education counties. According to research, systemic injustices and access barriers cause Black and Hispanic patients to have worse cancer outcomes [23]. According to a study by Zavala et al. (2021), the socioeconomic factors are associated with a higher mortality rate among Black patients with lung cancer [24]. This is given that the diversity of low-education counties coincides with populations dealing with exacerbated complications.

### 4.7. Particular Types of Cancer

The mortality rates (186.20–189.80) cover a range of cancers, and comparable cancer studies cause the differences. High smoking rates (19.65–20.08%) raise the incidence of lung cancer [25], and studies have found that low-education groups have higher rate ratios (3.01 for men and 2.02 for women) [21]. According to stage (33.8%) and co-morbidities, subsequent diagnoses decreased survival in a study on breast cancer [18]. Obesity and treatment delays are also factors in a colorectal cancer study [18]. Additionally, access barriers caused gaps in prostate and melanoma, with 50% of melanoma cases being critical [13]. Countries with primary preventative programs for cancers, such as cervical cancer, have seen lower rates of premature mortality [20].

### 4.8. Policy and Practice Implications

The results have important ramifications for screening initiatives, including colorectal screening units and mobile mammography, which could help with late diagnoses in rural areas (56.7–58.7% noncore) [26]. It is crucial to implement programs that target the diverse populations of low-education counties (7.09% Black, 5.31% Hispanic). With the help of telehealth, healthcare infrastructure should increase oncology services in low-SEP counties to decrease treatment delays [27]. The mortality increases (7.68 and 4.69) in the multivariate analysis emphasize the urgency. Given the high rates (19.65–20.08%) and the evidence that tobacco use contributes to disparities in lung cancer, behavioral interventions like smoking cessation campaigns are crucial [28]. Co-morbidities may be lessened by socioeconomic assistance that addresses poverty and food insecurity (16.30–16.70%) through subsidies [29]. Since low literacy is associated with negative beliefs about cancer [30], health literacy education programs in low-education counties could increase the screening uptake [31].

### 4.9. Strength and Weakness

Strength: We have employed a comprehensive dataset from AHRQ; data covering 3142 counties was one of the reliable real-world sources.

Limitations: Firstly, individual results cannot be inferred from a county-level analysis [32]. Undefined Thresholds: “Low employment” and “education” have no precise cut-offs, which makes them difficult to explain at the individual patient level and further the multivariate confounder adjustment that might be affected by residual confounding that is unaccounted for [33]. Cancer Specificity: Type-specific differences, such as lung versus breast, are concealed by aggregate mortality [34]. It is essential to acknowledge that our study has limitations, including a reliance solely on data from 2015 and the potential presence of unmeasured confounding variables, which may restrict the generalizability of our findings. Nonetheless, we expect that conducting further analyses, such as correlation analyses may prove challenging, as they could produce complex data, including non-linear or weak correlations, which are more difficult to interpret than binary comparisons. Additionally, there is a risk of overfitting, as the assumption of linearity in correlations is inherent; for example, the effects of unemployment may not be linear, as moderate unemployment might be less detrimental due to safety nets, thereby requiring meticulous model selection. Correlational analyses, similar to between-group analyses, involve county-level data, potentially leading to ecological fallacy (for example, presuming that county unemployment rates represent individual risks). These characteristics hinder the capacity to draw more robust correlations. We anticipate that forthcoming spatial disparity measurements related to economic opportunity, mobility, and health education will contribute to mitigating this issue. Given the limitations of our results, which stem from self-reported data, we anticipate that extensive findings from a county-level survey will prove advantageous. Moreover, exclusive dependence on education and work is insufficient to tackle socioeconomic inequities; therefore, other social determinants of health are crucial variables for comprehensive examination.

### 4.10. Future Research Implications

Additional research ought to update post-2020 data to evaluate the effects of COVID-19 [35], break it down by the type of cancer to identify the discrepancies [34], assess data at the individual level to support ecological conclusions [32], examine racial/ethnic interactions in light of the diversity shown in this study [23], and evaluate and assess the effectiveness of interventions (such as screening programs) in counties with low SEP [26].

#### 4.10.1. Current Implications of the Findings

Due to changes in healthcare, lifestyle, and policy regulations, the study’s findings from 2015 may not accurately reflect the potential changes in cancer mortality by 2025. Recent data from the CDC (2016–2020) indicate that the cancer mortality rates are declining more rapidly in urban areas (1.56–1.96% annually) compared to rural regions (0.93–1.43% annually). Nevertheless, rural regions exhibit elevated rates, with lung cancer mortality exceeding urban areas by 38% [36]. According to a 2024 study, the all-cause mortality rate among individuals in the low-income group increased by 3.3 times when compared to those in the high-income group [37]. According to GLOBOCAN 2022, the cancer mortality rates in the U.S. remain stable; however, low socioeconomic status populations continue to lag due to inadequate access to necessary care [38]. Our findings indicate that elevated smoking rates (19.65–20.08%) correlate with increased lung cancer mortality rates. This aligns with data from 2015 to 2019, indicating that the counties experiencing chronic poverty and rural regions exhibit elevated lung cancer rates associated with tobacco consumption [39].

A recent study indicates that low-socioeconomic position status (SEP) locations have elevated mortality rates for breast and colorectal cancer due to reduced screening participation, paralleling the rurality and health status depicted in our study [40]. The research indicated that the overall mortality rate in SEP counties ranged from 186.20 to 189.80 per 100,000, above the national average of 149.1 per 100,000 in 2019. Nevertheless, recent estimates from rural regions (160–170 per 100,000) indicate that these disparities persist [41]. The need for interventions—screening and access to healthcare—is much needed. Further, the healthcare decision making should take into consideration the progress made since 2015 (such as low-dose CT lung cancer screening) and ongoing problems in rural regions [41]. The more substantial effect for people with less education (7.68 vs. 4.69) is consistent with our understanding of how health literacy affects individuals [42].

#### 4.10.2. Cut-Off Low-Employment Status

Definitions of economic distress indicate that the threshold for the low-employment status likely corresponds to the upper quartile of unemployment rates (7–10%); however the rationale for this is not explicitly defined. The United States Department of Agriculture (USDA) Economic Research Services (ERS) employs comparable quartiles to identify at-risk counties, facilitating comparison [10]. The top quartile, comprising 906 counties, possesses a sufficiently substantial sample size for analysis; hence the standards are not excessively stringent. Elevated unemployment correlates with less access to insurance and treatment, directly impacting cancer outcomes [27]. The sensitivity of the cut-off to economic cycles remains ambiguous without a defined rate (e.g., 8% versus 10%), which may result in the misclassification of counties.

#### 4.10.3. Cut Offs for Low-Education Status

The USDA ERS regulations provide a criterion of ≥20% for persons lacking a high school diploma. This aligns with the government standards for poor educational achievement, facilitating comparisons [43]. A non-completion rate of 20% or above significantly indicates insufficient health literacy, a critical risk factor for cancer [21]. Additionally, focusing on rural Southern areas characterized by historical educational disparities, as illustrated in Table 1 (56.7% noncore). The binary cut-off oversimplifies the gradient effect of education, necessitating more nuanced thresholds to account for regional disparities, such as those between urban and rural areas.

#### 4.10.4. Internal and External Validity

The study’s total cancer death rate (186.20–189.80 per 100,000) is likely to include the most common types of cancer, although it might not be true for all. Our study shows that smoking rates are higher (19.65–20.08%), which suggests a strong link. The counties with low socioeconomic levels had higher lung cancer death rates due to an association with smoking [25]. Recent data from 2015 to 2019 show that the frequency is higher in the rural areas with poor socioeconomic status [7].

Among breast cancer, in low-socioeconomic areas, diagnoses happen later since there aren’t enough screens, as shown by the fact that they live in rural areas and are in poorer health conditions. This supports the studies that show differences in disease phases [19]. The multivariate results (7.68 and 4.69) are likely to include the effects of breast cancer. Additionally, food insecurity (16.30–16.70%) and poor health lead to obesity and less screening, which raises the risk of death [22]. Moreover, new studies suggest that access to care may be a crucial factor in cancer outcomes [44].

#### 4.10.5. Internal Validity

Strength and positive aspects are mostly derived from the robustness of the data. Utilizing data from the AHRQ that links information from the CDC, Census, and ERS ensures the precision of indicators such as mortality rates and smoking prevalence. Controlling for covariates such as smoking and rural residency isolates the impacts of the socioeconomic position, seen in the substantial results (7.68 and 4.69) [45]. The extensive sample of 3142 counties enhances the study’s statistical power and diminishes random error (*p* < 0.001).

Weakness:

The ecological fallacy: County-level statistics cannot be utilized to ascertain effects on individuals, perhaps resulting in erroneous findings regarding relationships [46].

Confounding: Omitted variables, such as migration or the cancer stage, can introduce bias due to unreported changes [5].

Cross-sectional design: The causal relationships remain ambiguous due to the lack of a clear temporal framework [34].

Inadequate proxies: Home healthcare and urban-rural variables inadequately assess access, thereby diminishing observed effects [14,47]. Thus, internal validity is moderate due to robust data and adjustments. Readers must exercise caution, since the findings should be interpreted with an awareness of the constraints inherent in ecological design and the use of proxies.

#### 4.10.6. External Validity

Strength and positive aspects arise from the national scope of the findings. The 3142 counties in the U.S. cover a wide range of situations, which makes them more useful [31]. The findings align with prior research on cancer mortality in rural areas and low socioeconomic status areas, hence enhancing its generalizability [48,49]. The findings are applicable to other high-income countries exhibiting rural/SEP disparities [50].

Weakness: Trends from 2015 may differ significantly from those in 2025 (e.g., advancements in telemedicine and screening), complicating temporal generalizations [51].

Specific to the United States: The findings may not be applicable to low- and middle-income countries with diverse healthcare systems [32]. County-Level Focus: It does not consider individual-level factors, such as patient beliefs, and hence cannot be utilized for individualized therapies [46]. Regional Bias: Low-socioeconomic groups predominantly originate from the South and rural regions (Table 1), thereby diminishing the applicability of the results nationwide [33]. Thus the external validity is moderate; it is robust for rural and low-SEP regions in the U.S. but weakened by the data’s age and ecological boundaries.

#### 4.10.7. Additional Contributing Factors

Moreover, diminished access to education and employment opportunities may lead to lifestyle factors (such as suboptimal diets, insufficient sun exposure, and elevated rates of obesity and diabetes) that correlate with heightened cancer risk and incidence, as indicated in the solar UVB study [52]. Furthermore, metabolism, hormone secretion, time zones [53] are additional factors that have implications. Consequently, socioeconomic disparities likely affect cancer outcomes both directly and indirectly through modifiable risk factors.

## 5. Conclusions

Low-employment and low-education counties experience notable differences in cancer mortality, with increases of 4.69 and 7.68 per 100,000, respectively, according to the results of the unadjusted and adjusted multivariate analyses. Rurality, high smoking rates, poor health, food insecurity, and systemic barriers—all of which are exacerbated by racial and ethnic diversity—are the leading causes of these. In line with the identified gaps, recent research emphasizes co-morbidities, treatment delays, and late diagnoses as important mechanisms. To lessen these disparities, targeted interventions that address socioeconomic issues, behaviors, and access are necessary, with an emphasis on the vital role that education plays.

## Figures and Tables

**Figure 1 cancers-17-02051-f001:**
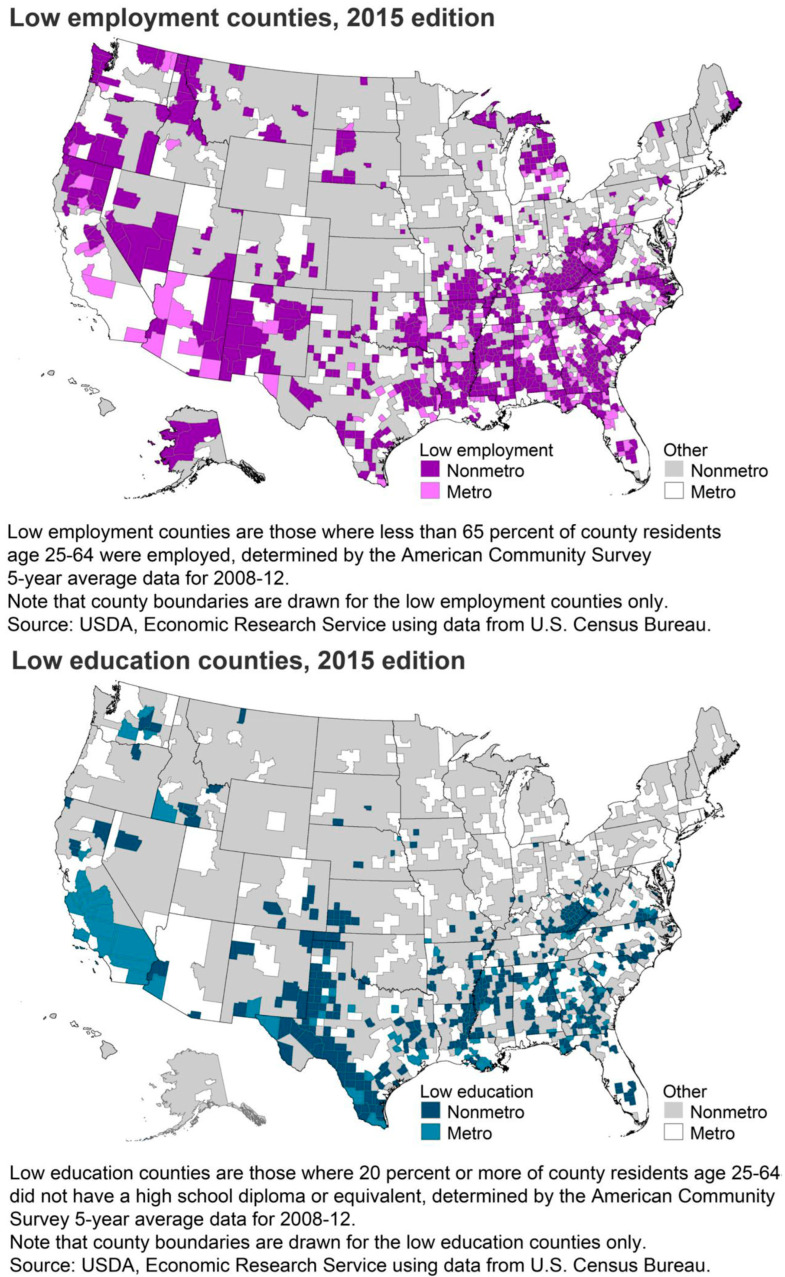
Low employment and education counties in the United States.

**Table 1 cancers-17-02051-t001:** Baseline characteristics of counties with and without low-education and low-employment status in Year 2015.

	Counties Without Low-Employment Status	Counties with Low-Employment Status	*p*-Value	Counties Without Low-Education Status	Counties with Low-Education Status	*p*-Value
**n**	2236 (71.16%)	906 (28.84%)		2675 (85.14%)	467 (14.86%)	
**Sex**			<0.001			<0.001
Percentage of Female (median [IQR])	50.46 [49.70, 51.11]	50.36 [48.31, 51.22]		50.47 [49.64, 51.14]	50.08 [47.92, 51.08]	
**Race**						
Percentage of White (median [IQR])	91.40 [81.45, 95.69]	84.43 [63.57, 94.25]	<0.001	91.14 [80.04, 95.70]	78.97 [61.20, 91.20]	<0.001
Percentage of Black (median [IQR])	1.57 [0.54, 6.76]	5.28 [0.96, 29.09]	<0.001	1.81 [0.56, 8.30]	7.09 [1.42, 30.54]	<0.001
Percentage of Hispanic (median [IQR])	4.19 [2.14, 9.78]	2.82 [1.52, 6.22]	<0.001	3.56 [1.88, 7.81]	5.31 [1.92, 34.05]	<0.001
Percentage of Asian (median [IQR]))	0.67 [0.32, 1.54]	0.35 [0.13, 0.65]	<0.001	0.59 [0.28, 1.31]	0.35 [0.11, 0.74]	<0.001
Percentage of Alaska Indian or American Indians (median [IQR])	0.34 [0.17, 0.76]	0.30 [0.11, 0.86]	<0.001	0.34 [0.16, 0.79]	0.29 [0.09, 0.75]	<0.001
Percentage of Others (median [IQR])	0.91 [0.34, 2.38]	0.64 [0.19, 1.73]	<0.001	0.75 [0.27, 1.90]	1.65 [0.46, 5.53]	<0.001
**NCHS Urban-Rural Code (%)**			<0.001			<0.001
Large Central Metropolitan & Large Fringe Metropolitan	400 (17.9)	36 (4.0)		413 (15.4)	23 (4.9)	
Medium Metropolitan	306 (13.7)	66 (7.3)		334 (12.5)	38 (8.1)	
Small Metropolitan	274 (12.3)	84 (9.3)		319 (11.9)	39 (8.4)	
Micropolitan	453 (20.3)	188 (20.8)		539 (20.1)	102 (21.8)	
Noncore	803 (35.9)	532 (58.7)		1070 (40.0)	265 (56.7)	
**Healthcare Services**			<0.001			<0.001
Percentage of home healthcare services per 1000 people (median [IQR])	0.08 [0.05, 0.13]	0.11 [0.06, 0.16]		0.09 [0.06, 0.13]	0.10 [0.06, 0.17]	
**Smoking**			<0.001			<0.001
Percentage of adults who are current smokers (median [IQR])	16.52 [15.05, 18.48]	20.08 [17.93, 22.71]		17.06 [15.30, 19.47]	19.65 [17.01, 22.47]	
**Health Status**						
Percentage of adults reporting fair or poor health (age-adjusted) (median [IQR])	14.61 [12.66, 17.64]	20.77 [17.76, 23.66]	<0.001	15.14 [13.02, 18.52]	22.46 [19.52, 25.66]	<0.001
**Food Access**						
Percentage of population who lack adequate access to food (median [IQR])	12.60 [10.70, 14.60]	16.70 [14.50, 19.70]	<0.001	13.40 [11.30, 15.80]	16.30 [12.60, 19.85]	<0.001
**Drinking**						
Percentage of adults reporting binge or heavy drinking (median [IQR])	17.47 [15.45, 19.26]	14.12 [12.26, 16.47]	<0.001	17.07 [14.82, 19.06]	14.53 [12.38, 16.40]	<0.001
**Cancer Mortality**						
**Total age-adjusted cancer mortality per 100,000 people in the county** (median [IQR])	169.15 [154.00, 183.50]	189.80 [171.90, 207.10]	<0.001	172.90 [157.00, 188.40]	186.20 [161.72, 209.33]	<0.001

Abbreviations: IQR, Interquartile range; NCHS, National Center for Health Statistics.

## Data Availability

We used publicly available Social determinants of health data from the Agency for Healthcare Quality (AHRQ) 2015 county database.

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
