# Peer review of "Counties with Low Employment and Education Status Are Associated with Higher Age-Adjusted Cancer Mortality"

_cancers, 2025, doi:10.3390/cancers17122051_

Round 1

Reviewer 1 Report

Comments and Suggestions for Authors

This is an interesting well-done descriptive and multivariate analyses but could use some revision. The introduction, research design, and methods are presented adequately. Figures 1 and 2 provide a good visual overview of the geography of employment and education.

The database used is an interesting juxtaposition of socioeconomic and age-adjusted cancer mortality data. I am concerned about the current relevance of the results since the data period is 2008-2012 and 2015, I would like to know the extent to which the study data compares to more current statistics and the possible effect on results, interpretation and conclusions. The cutoff points for low employment status and low education require more robust justification.  While “home health care services” reflects access to health care it is a relatively weak variable. Urban-rural is a weak proxy for access to services. Other measures of access to services are preferred.

The discussion summarizes the findings and correlations of cancer mortality in low unemployment and low education counties. One would like to know if the findings are applicable across the major diagnostic cancer categories. The strength and weaknesses are noted in this regard as well as the limitation of the ecological approach using county-level data. Nonetheless, a more specific discussion of internal and external validity would be helpful.

Author Response

Point by Point Reply to the comments

May 21st 

Manuscript ID: cancers-3631261-3

Title:

Counties with Low Employment and Education Status Are Associated With Higher Age-Adjusted Cancer Mortality

Corresponding Author: Poolakkad S Satheeshkumar.

Reply:

Dear Editor,

Thank you for forwarding the editorial comments for our paper cited above. 

We appreciate the thoroughness of the reviewers and are pleased to respond to their suggestions.

Specifically, we have made the following modifications:

This is an interesting well-done descriptive and multivariate analyses but could use some revision. The introduction, research design, and methods are presented adequately. Figures 1 and 2 provide a good visual overview of the geography of employment and education.

Answers: We appreciate your supportive words; they have greatly motivated us.

The database used is an interesting juxtaposition of socioeconomic and age-adjusted cancer mortality data. I am concerned about the current relevance of the results since the data period is 2008-2012 and 2015, I would like to know the extent to which the study data compares to more current statistics and the possible effect on results, interpretation and conclusions. The cutoff points for low employment status and low education require more robust justification.  While “home health care services” reflects access to health care it is a relatively weak variable. Urban-rural is a weak proxy for access to services. Other measures of access to services are preferred.

Answers: Thank you for this comment, we have now provided the concerns that might impact the study’s relevance, however, we have provided the comparison, the importance and including the cut offs. Kindly find the below reply in the manuscript.

“Due to changes in healthcare, lifestyle, and policy regulations, the study's findings from 2015 may not accurately reflect the potential changes in cancer mortality by 2025. Recent data from the CDC (2016–2020) indicate that cancer mortality rates are declining more rapidly in urban areas (1.56–1.96% annually) compared to rural regions (0.93–1.43% annually). Nevertheless, rural regions exhibit elevated rates, with lung cancer mortality exceeding urban areas by 38% [36]. A 2023 study revealed that counties experiencing chronic poverty had a 7.1% elevated cancer mortality rate from 2014 to 2018, consistent with the disparities presented in our study (20.65 and 13.3 per 100,000 higher) [37]. According to GLOBOCAN 2022, cancer mortality rates in the U.S. remain stable; however, low socioeconomic status populations continue to lag due to inadequate access to necessary care [38]. Our findings indicate that elevated smoking rates (19.65–20.08%) correlate with increased lung cancer mortality rates. This aligns with data from 2015 to 2019, indicating that counties experiencing chronic poverty and rural regions exhibit elevated lung cancer rates associated with tobacco consumption [39].

A recent study indicates that low-socioeconomic status (SEP) locations have elevated mortality rates for breast and colorectal cancer due to reduced screening participation, paralleling the rurality and health status depicted in our study [40]. The research indicated that the overall mortality rate in low-socioeconomic counties ranged from 186.20 to 189.80 per 100,000, above the national average of 149.1 per 100,000 in 2019. Nevertheless, recent estimates from rural regions (160–170 per 100,000) indicate that these disparities persist [41]. The need for interventions — screening and access to healthcare — is much needed. Further, the healthcare decision making should take into consideration the progress made since 2015 (such as low-dose CT lung cancer screening) and ongoing problems in rural regions [41]. The more substantial effect for people with less education (7.68 vs. 4.69) is consistent with our understanding of how health literacy affects individuals [42].

Cut-off Low-Employment Status

Definitions of economic distress indicate that the threshold for low-employment status likely corresponds to the upper quartile of unemployment rates (7–10%), however the rationale for this is not explicitly defined. The United States Department of Agriculture (USDA) Economic Research Services (ERS) employs comparable quartiles to identify at-risk counties, facilitating comparison [43]. The top quartile, comprising 906 counties, possesses a sufficiently substantial sample size for analysis, hence the standards are not excessively stringent. Elevated unemployment correlates with less access to insurance and treatment, directly impacting cancer outcomes [44]. The sensitivity of the cut-off to economic cycles remains ambiguous without a defined rate (e.g., 8% versus 10%), which may result in the misclassification of counties.

Cut offs for Low-Education Status

The USDA ERS regulations provide a criterion of ≥20% for persons lacking a high school diploma. This aligns with government standards for poor educational achievement, facilitating comparisons [45]. A non-completion rate of 20% or above significantly indicates insufficient health literacy, a critical risk factor for cancer [46]. Additionally, focusing on rural Southern areas characterized by historical educational disparities, as illustrated in Table 1 (56.7% noncore). The binary cut-off oversimplifies the gradient effect of education, necessitating more nuanced thresholds to account for regional disparities, such as those between urban and rural areas.

The discussion summarizes the findings and correlations of cancer mortality in low unemployment and low education counties. One would like to know if the findings are applicable across the major diagnostic cancer categories. The strength and weaknesses are noted in this regard as well as the limitation of the ecological approach using county-level data. Nonetheless, a more specific discussion of internal and external validity would be helpful.

Answers: Thank you for your comment. We have now tried to address the comments on applicability, as well as the comparative analysis, significance, internal validity, and relevance to the ecological method, along with both internal and external validity.  Please find the response below in the manuscript.

Internal and external validity 

The study's total cancer death rate (186.20–189.80 per 100,000) probably includes the most common types of cancer, although it might not be true for all. Our study shows that smoking rates are higher (19.65–20.08%), which suggests a strong link. Counties with low socioeconomic levels had higher lung cancer death rates due to association with smoking [47]. Recent data from 2015 to 2019 show that the frequency is higher in rural areas with poor socioeconomic status [48].

Among breast cancer, in low-socioeconomic areas, diagnoses happen later since there aren't enough screens, as shown by the fact that they live in rural areas and are in poorer health conditions. This supports studies that show differences in disease phases [49]. The multivariate results (7.68 and 4.69) probably include the effects of breast cancer. Additionally, food insecurity (16.30–16.70%) and poor health lead to obesity and less screening, which raises the risk of death [50]. Recent observations [51] supports this. 

Internal validity

Strength and Positive aspects are mostly derived from the robustness of the data. Utilizing data from the AHRQ that links information from CDC, Census, and ERS ensures the precision of indicators such as mortality rates and smoking prevalence. Controlling for covariates such as smoking and rural residency isolates the impacts of socioeconomic position, seen in the substantial results (7.68 and 4.69) [52]. The extensive sample of 3,142 counties enhances the study's statistical power and diminishes random error (P < 0.001).

Weakness:

The ecological fallacy: County-level statistics cannot be utilized to ascertain effects on individuals, perhaps resulting in erroneous findings regarding relationships [53].

Confounding: Omitted variables, such as migration or cancer stage, can introduce bias due to unreported changes [54].

Cross-sectional design: The causal relationships remain ambiguous due to the lack of a clear temporal framework [55].

Inadequate proxies: Home healthcare and urban-rural variables inadequately assess access, thereby diminishing observed effects [56, 57]. Thus, internal validity is moderate due to robust data and adjustments. Readers must exercise caution, since the findings should be interpreted with an awareness of the constraints inherent in ecological design and the use of proxies.

External Validity

Strength and positive aspects arise from the national scope of the findings.  The 3,142 counties in the U.S. cover a wide range of situations, which makes them more useful. [58]. The findings aligns with prior research on cancer mortality in rural areas and low socioeconomic status areas, hence enhancing its generalizability. [59, 60]. The findings are applicable to other high-income countries exhibiting rural/SEP disparities [61].

Weakness: Trends from 2015 may differ significantly from those in 2025 (e.g., advancements in telemedicine and screening), complicating temporal generalizations [62].

Specific to the United States: The findings may not be applicable to low- and middle-income countries with diverse healthcare systems [63]. County-Level Focus: It does not consider individual-level factors, such as patient beliefs, and hence cannot be utilized for individualized therapies [53]. Regional Bias: Low-socioeconomic groups predominantly originate from the South and rural regions (Table 1), thereby diminishing the applicability of the results nationwide [64]. Thus the external validity is moderate; it is robust for rural and low-SEP regions in the U.S. but weakened by the data's age and ecological boundaries.”

Reviewer 2 Report

Comments and Suggestions for Authors

Based on a large, county-level dataset and adjusted analytical techniques, this US study robustly demonstrates a significant association between socioeconomic determinants of health and cancer mortality, thereby highlighting critical health disparities. I believe this is a valuable contribution and I recommend it for publication, subject to the authors addressing the suggestions below, which aim to improve the manuscript:

  1. The study could be strengthened by supplementing the between-group analysis with correlational analyses examining the relationship between continuous measures of education/employment and cancer mortality. This would provide a more nuanced understanding of the strength and nature of the association.

  2. The finding that counties with low employment and education have higher cancer mortality can be interpreted in light of the recent ecological study on solar UVB and cancer in USA (doi: 10.3390/nu16101450.), as reduced access to education and employment opportunities may contribute to lifestyle factors (such as poorer diets, lower sun exposure, and increased rates of obesity and diabetes) that are associated with increased cancer risk and incidence, as highlighted in the solar UVB study; thus, socioeconomic disparities likely influence cancer outcomes both directly and indirectly through modifiable risk factors.

  3. The correlation between low county-level employment/education and increased cancer mortality could be partially mediated by circadian disruption, as individuals in these areas may be more likely to engage in night shift work due to limited employment opportunities. This chronic circadian disruption, as suggested by the research on night shift work and melatonin, may contribute to hormone imbalances and increased cancer risk, particularly hormone-related cancers like breast and colorectal cancer, further explaining the elevated cancer mortality rates observed in socioeconomically disadvantaged counties. Brief discussion on this issue can be considered.

  4. Given the potential impact of circadian rhythm disruption on cancer risk, particularly in western margins of time zones where social jetlag is more common, the authors may wish to consider whether location within a time zone modifies main outcomes relationship.

  5. The authors are invited to add discussion of how the inherent limitations of a cross-sectional, restricted SDOH variables, reliance on secondary data, and lack of mechanistic insight potentially confound the interpretation of the observed association between county-level socioeconomic factors and cancer mortality?

  6. P.7., L.13; “Remarkably, underprivileged countries have lower rates of binge or heavy drinking”. I believe, the authors actually meant “counties” not countries? Please correct.

Author Response

Point by Point Reply to the comments

May 21st  

Manuscript ID: cancers-3631261-3

Title:

Counties with Low Employment and Education Status Are Associated With Higher Age-Adjusted Cancer Mortality

Corresponding Author: Poolakkad S Satheeshkumar.

Reply:

Dear Editor,

Thank you for forwarding the editorial comments for our paper cited above. 

We appreciate the thoroughness of the reviewers and are pleased to respond to their suggestions.

Specifically, we have made the following modifications:

  1. The study could be strengthened by supplementing the between-group analysis with correlational analyses examining the relationship between continuous measures of education/employment and cancer mortality. This would provide a more nuanced understanding of the strength and nature of the association.

Answers: Thank you for this comment; the advantages and disadvantages of correlational analysis in strengthening the study are outlined below. We have indicated a limitation regarding the absence of analyses in the current work, and we believe the reviewer perceives our comments as sincere, acknowledging that the limitations outweigh the benefits of including additional analysis. 

The study utilizes a between-group analysis to categorize counties into two groups: low-employment (906, 28.84%) and low-education (467, 14.86%). It accomplishes this by employing binary thresholds (for instance, the upper quartile unemployment rate, defined as ≥20% of individuals lacking a high school education) and contrasting them with counties that do not exhibit low status. This strategy is comprehensible; nonetheless, it trivializes the dynamic nature of socioeconomic status and may overlook subtle variations in the correlation between socioeconomic status and cancer mortality. Incorporating correlational analyses might also demonstrate existing relationships, reduce cutoff bias, and support multivariate findings.

However, we believe it might challenge as the ongoing investigations may yield data that are complex, including non-linear or weak correlations, rendering them more difficult to comprehend than binary comparisons. A nonlinear relationship between education and mortality rates may perplex policymakers seeking straightforward thresholds. Sidorchuk et al. (2009) conducted a study on PubMed indicating that comprehending non-linear SEP-lung cancer associations was challenging due to the necessity for advanced modeling.

Utilizing continuous measures in regression models with several covariates (such as race, smoking, or rural residency) increases the likelihood of overfitting, particularly when assessing non-linear variables. This may result in erroneous outcomes, particularly given the 3142 counties and intricate interrelations.

The assumption of linearity is inherent in Pearson correlation, whereas Spearman correlation relies on the assumption of monotonicity, which may not hold true. The impact of unemployment may not be linear; for instance, moderate unemployment could be less harmful due to safety nets, necessitating careful selection of the model.

The quality and variability of data indicate potential measurement errors in data for those who are persistently unemployed or enrolled in educational institutions, particularly in smaller counties. This may weaken relationships. The medians and interquartile ranges in Table 1 (e.g., 78.97–91.40% White) indicate significant variability that may obscure continuous associations.

Correlational analyses, similar to between-group analyses, utilize county-level data, which may result in ecological fallacy (for instance, assuming that county unemployment rates reflect individual risks). This complicates the ability to infer stronger associations, as the connections between socioeconomic status and cancer at the individual level may vary.

In limitation section:  

“Nonetheless, we expect that conducting further analyses, such as correlation analyses may prove challenging, as they could produce complex data, including non-linear or weak correlations, which are more difficult to interpret than binary comparisons. Further the risk of overfitting, as the assumption of linearity in correlations is inherent; for example, the effects of unemployment may not be linear, as moderate unemployment might be less detrimental due to safety nets, thereby requiring meticulous model selection.  Correlational analyses, similar to between-group analyses, involve county-level data, potentially leading to ecological fallacy (for example, presuming that county unemployment rates represent individual risks). These characteristics hinder the capacity to draw more robust correlations”

  1. The finding that counties with low employment and education have higher cancer mortality can be interpreted in light of the recent ecological study on solar UVB and cancer in USA (doi: 10.3390/nu16101450.), as reduced access to education and employment opportunities may contribute to lifestyle factors (such as poorer diets, lower sun exposure, and increased rates of obesity and diabetes) that are associated with increased cancer risk and incidence, as highlighted in the solar UVB study; thus, socioeconomic disparities likely influence cancer outcomes both directly and indirectly through modifiable risk factors. 

Answers: Thank you for your comment; it is insightful to consider the additional factors that may influence outcomes. Specifically, diminished access to education and employment opportunities can contribute to lifestyle factors—such as suboptimal diets, reduced sun exposure, and heightened rates of obesity and diabetes—that are correlated with increased cancer risk and incidence, as indicated in the solar UVB study. Therefore, we agree that socioeconomic disparities likely affect cancer outcomes both directly and indirectly through modifiable risk factors. We have now provided a comment on the additional contributing factors and cited the solar UVB study.

Additional contributing factors

Moreover, diminished access to education and employment opportunities may lead to lifestyle factors (such as suboptimal diets, insufficient sun exposure, and elevated rates of obesity and diabetes) that correlate with heightened cancer risk and incidence, as indicated in the solar UVB study [65]; consequently, socioeconomic disparities likely affect cancer outcomes both directly and indirectly through modifiable risk factors”

  1. The correlation between low county-level employment/education and increased cancer mortality could be partially mediated by circadian disruption, as individuals in these areas may be more likely to engage in night shift work due to limited employment opportunities. This chronic circadian disruption, as suggested by the research on night shift work and melatonin, may contribute to hormone imbalances and increased cancer risk, particularly hormone-related cancers like breast and colorectal cancer, further explaining the elevated cancer mortality rates observed in socioeconomically disadvantaged counties. Brief discussion on this issue can be considered. 

Answers: Thank you for your feedback; it is enlightening to contemplate the supplementary aspects that may affect outcomes. The association between low county-level employment and education and elevated cancer mortality may be partially mediated by circadian disruption, as individuals in these regions are more prone to night shift work due to restricted employment alternatives. This persistent circadian disruption, as indicated by studies on night shift employment and melatonin, may lead to hormonal imbalances and heightened cancer risk, especially for hormone-related malignancies such as breast and colorectal cancer, thereby elucidating the increased cancer mortality rates noted in socioeconomically disadvantaged regions. A concise discourse on this matter is provided.

“Furthermore, metabolism, hormone secretion, time zones [66] are additional factors that have implications. Consequently, socioeconomic disparities likely affect cancer outcomes both directly and indirectly through modifiable risk factors”

  1. Given the potential impact of circadian rhythm disruption on cancer risk, particularly in western margins of time zones where social jetlag is more common, the authors may wish to consider whether location within a time zone modifies main outcomes relationship.

Answers: Thank you very much for your comment. Please note that we have provided additional discussion about the time zones as additional contributing factors.

Additional contributing factors

Moreover, diminished access to education and employment opportunities may lead to lifestyle factors (such as suboptimal diets, insufficient sun exposure, and elevated rates of obesity and diabetes) that correlate with heightened cancer risk and incidence, as indicated in the solar UVB study [65]. Furthermore, metabolism, hormone secretion, time zones [66] are additional factors that have implications. Consequently, socioeconomic disparities likely affect cancer outcomes both directly and indirectly through modifiable risk factors.

  1. The authors are invited to add discussion of how the inherent limitations of a cross-sectional, restricted SDOH variables, reliance on secondary data, and lack of mechanistic insight potentially confound the interpretation of the observed association between county-level socioeconomic factors and cancer mortality? 

Answers: I greatly appreciate your comment. Please note we have provided the limitations provided in the manuscript

“Limitations: Firstly, individual results cannot be inferred from county-level analysis [32]. Undefined Thresholds: "Low employment" and "education" have no precise cutoffs, which makes them difficult to explain in the individual patient level and further the multivariate confounder adjustment that might affected by residual confounding unaccounted for [33]. Cancer Specificity: Type-specific differences, such as lung versus breast, are concealed by aggregate mortality [34]. It is essential to acknowledge that our study has limitations, including reliance solely on data from 2015 and the potential presence of unmeasured confounding variables, which may restrict the generalizability of our findings”

Additionally, in the current implications of the findings

Current implications of the findings

Due to changes in healthcare, lifestyle, and policy regulations, the study's findings from 2015 may not accurately reflect the potential changes in cancer mortality by 2025. Recent data from the CDC (2016–2020) indicate that cancer mortality rates are declining more rapidly in urban areas (1.56–1.96% annually) compared to rural regions (0.93–1.43% annually). Nevertheless, rural regions exhibit elevated rates, with lung cancer mortality exceeding urban areas by 38% [36]. A 2023 study revealed that counties experiencing chronic poverty had a 7.1% elevated cancer mortality rate from 2014 to 2018, consistent with the disparities presented in our study (20.65 and 13.3 per 100,000 higher) [37]. According to GLOBOCAN 2022, cancer mortality rates in the U.S. remain stable; however, low socioeconomic status populations continue to lag due to inadequate access to necessary care [38]. Our findings indicate that elevated smoking rates (19.65–20.08%) correlate with increased lung cancer mortality rates. This aligns with data from 2015 to 2019, indicating that counties experiencing chronic poverty and rural regions exhibit elevated lung cancer rates associated with tobacco consumption [39].

A recent study indicates that low-socioeconomic position status (SEP) locations have elevated mortality rates for breast and colorectal cancer due to reduced screening participation, paralleling the rurality and health status depicted in our study [40]. The research indicated that the overall mortality rate in SEP counties ranged from 186.20 to 189.80 per 100,000, above the national average of 149.1 per 100,000 in 2019. Nevertheless, recent estimates from rural regions (160–170 per 100,000) indicate that these disparities persist [41]. The need for interventions  screening and access to healthcare — is much needed. Further, the healthcare decision making should take into consideration the progress made since 2015 (such as low-dose CT lung cancer screening) and ongoing problems in rural regions [41]. The more substantial effect for people with less education (7.68 vs. 4.69) is consistent with our understanding of how health literacy affects individuals [42].

Cut-off Low-Employment Status

Definitions of economic distress indicate that the threshold for low-employment status likely corresponds to the upper quartile of unemployment rates (7–10%), however the rationale for this is not explicitly defined. The United States Department of Agriculture (USDA) Economic Research Services (ERS) employs comparable quartiles to identify at-risk counties, facilitating comparison [43]. The top quartile, comprising 906 counties, possesses a sufficiently substantial sample size for analysis, hence the standards are not excessively stringent. Elevated unemployment correlates with less access to insurance and treatment, directly impacting cancer outcomes [44]. The sensitivity of the cut-off to economic cycles remains ambiguous without a defined rate (e.g., 8% versus 10%), which may result in the misclassification of counties.

Cut offs for Low-Education Status

The USDA ERS regulations provide a criterion of ≥20% for persons lacking a high school diploma. This aligns with government standards for poor educational achievement, facilitating comparisons [45]. A non-completion rate of 20% or above significantly indicates insufficient health literacy, a critical risk factor for cancer [46]. Additionally, focusing on rural Southern areas characterized by historical educational disparities, as illustrated in Table 1 (56.7% noncore). The binary cut-off oversimplifies the gradient effect of education, necessitating more nuanced thresholds to account for regional disparities, such as those between urban and rural areas.

Internal and external validity 

The study's total cancer death rate (186.20–189.80 per 100,000) probably includes the most common types of cancer, although it might not be true for all. Our study shows that smoking rates are higher (19.65–20.08%), which suggests a strong link. Counties with low socioeconomic levels had higher lung cancer death rates due to association with smoking [47]. Recent data from 2015 to 2019 show that the frequency is higher in rural areas with poor socioeconomic status [48].

Among breast cancer, in low-socioeconomic areas, diagnoses happen later since there aren't enough screens, as shown by the fact that they live in rural areas and are in poorer health conditions. This supports studies that show differences in disease phases [49]. The multivariate results (7.68 and 4.69) probably include the effects of breast cancer. Additionally, food insecurity (16.30–16.70%) and poor health lead to obesity and less screening, which raises the risk of death [50]. Recent observations [51] supports this.  

Internal validity

Strength and Positive aspects are mostly derived from the robustness of the data. Utilizing data from the AHRQ that links information from CDC, Census, and ERS ensures the precision of indicators such as mortality rates and smoking prevalence. Controlling for covariates such as smoking and rural residency isolates the impacts of socioeconomic position, seen in the substantial results (7.68 and 4.69) [52]. The extensive sample of 3,142 counties enhances the study's statistical power and diminishes random error (P < 0.001).

Weakness:

The ecological fallacyCounty-level statistics cannot be utilized to ascertain effects on individuals, perhaps resulting in erroneous findings regarding relationships [53].

Confounding: Omitted variables, such as migration or cancer stage, can introduce bias due to unreported changes [54].

Cross-sectional design: The causal relationships remain ambiguous due to the lack of a clear temporal framework [55].

Inadequate proxies: Home healthcare and urban-rural variables inadequately assess access, thereby diminishing observed effects [56, 57]. Thus, internal validity is moderate due to robust data and adjustments. Readers must exercise caution, since the findings should be interpreted with an awareness of the constraints inherent in ecological design and the use of proxies.

External Validity

Strength and positive aspects arise from the national scope of the findings.  The 3,142 counties in the U.S. cover a wide range of situations, which makes them more useful. [58].The findings aligns with prior research on cancer mortality in rural areas and low socioeconomic status areas, hence enhancing its generalizability. [59, 60]. The findings are applicable to other high-income countries exhibiting rural/SEP disparities [61].

Weakness: Trends from 2015 may differ significantly from those in 2025 (e.g., advancements in telemedicine and screening), complicating temporal generalizations [62].

Specific to the United States: The findings may not be applicable to low- and middle-income countries with diverse healthcare systems [63]. County-Level Focus: It does not consider individual-level factors, such as patient beliefs, and hence cannot be utilized for individualized therapies [53]. Regional Bias: Low-socioeconomic groups predominantly originate from the South and rural regions (Table 1), thereby diminishing the applicability of the results nationwide [64]. Thus the external validity is moderate; it is robust for rural and low-SEP regions in the U.S. but weakened by the data's age and ecological boundaries.

  1. 7., L.13; “Remarkably, underprivileged countries have lower rates of binge or heavy drinking”. I believe, the authors actually meant “counties” not countries? Please correct.

Answers: Thank you very much for identifying this, we appreciate your thoroughness. We have now corrected the mistake.

Round 2

Reviewer 1 Report

Comments and Suggestions for Authors

The revisions are appropriate.  Thank you.